# An Innovative Scoring System to Select the Optimal Surgery in Breast Cancer after Neoadjuvant Chemotherapy

**DOI:** 10.3390/jpm13081280

**Published:** 2023-08-21

**Authors:** Antonio Franco, Alba Di Leone, Marco Conti, Alessandra Fabi, Luisa Carbognin, Andreina Daniela Terribile, Paolo Belli, Armando Orlandi, Martin Alejandro Sanchez, Francesca Moschella, Elena Jane Mason, Giovanni Cimino, Alessandra De Filippis, Fabio Marazzi, Ida Paris, Giuseppe Visconti, Liliana Barone Adesi, Lorenzo Scardina, Sabatino D’Archi, Marzia Salgarello, Diana Giannarelli, Riccardo Masetti, Gianluca Franceschini

**Affiliations:** 1Breast Unit, Department of Women, Children and Public Health Sciences, Fondazione Policlinico Universitario “A. Gemelli” IRCCS, 00168 Roma, Italy; antonio.franco@guest.policlinicogemelli.it (A.F.);; 2Diagnostic Radiology and General Interventional Radiology, Fondazione Policlinico Universitario “A. Gemelli” IRCCS, 00168 Roma, Italy; 3Precision Medicine Breast Unit, Scientific Directorate, Department of Women, Children and Public Health Sciences, Fondazione Policlinico Universitario “A. Gemelli” IRCCS, 00168 Roma, Italy; 4Department of Women, Children and Public Health Sciences, Fondazione Policlinico Universitario “A. Gemelli” IRCCS, 00168 Roma, Italy; 5Medical Oncology, Department of Medical and Surgical Sciences, Fondazione Policlinico Universitario “A. Gemelli” IRCCS, 00168 Roma, Italy; 6Cancer Radiation Therapy, Department of Diagnostic Imaging, Oncological Radiotherapy and Hematology, Fondazione Policlinico Universitario “A. Gemelli” IRCCS, 00168 Roma, Italy; 7Plastic Surgery, Department of Women, Children and Public Health Sciences, Fondazione Policlinico Universitario “A. Gemelli” IRCCS, 00168 Roma, Italy; 8Epidemiology and Biostatistics, Fondazione Policlinico Universitario “A. Gemelli” IRCCS, 00168 Roma, Italy

**Keywords:** breast cancer, neoadjuvant treatment, surgery, scoring system, breast-conserving surgery, oncoplastic techniques, conservative mastectomy, personalized medicine, aesthetic outcomes, oncological outcomes

## Abstract

Introduction: The selection of surgery post-neoadjuvant chemotherapy (NACT) is difficult and based on surgeons’ expertise. The aim of this study was to create a post-NEoadjuvant Score System (pNESSy) to choose surgery, optimizing oncological and aesthetical outcomes. Methods: Patients (stage I–III) underwent surgery post-NACT (breast-conserving surgery (BCS), oncoplastic surgery (OPS), and conservative mastectomy (CMR) were included. Data selected were BRCA mutation, ptosis, breast volume, radiological response, MRI, and mammography pre- and post-NACT prediction of excised breast area. pNESSy was created using the association between these data and surgery. Area under the curve (AUC) was assessed. Patients were divided into groups according to correspondence (G1) or discrepancy (G2) between score and surgery; oncological and aesthetic outcomes were analyzed. Results: A total of 255 patients were included (118 BCS, 49 OPS, 88 CMR). pNESSy between 6.896–8.724 was predictive for BCS, 8.725–9.375 for OPS, and 9.376–14.245 for CMR; AUC was, respectively, 0.835, 0.766, and 0.825. G1 presented a lower incidence of involved margins (5–14.7%; *p* = 0.010), a better locoregional disease-free survival (98.8–88.9%; *p* < 0.001) and a better overall survival (96.1–86.5%; *p* = 0.017), and a better satisfaction with breasts (39.8–27.5%; *p* = 0.017) and physical wellbeing (93.5–73.6%; *p* = 0.001). Conclusion: A score system based on clinical and radiological features was created to select the optimal surgery post-NACT and improve oncological and aesthetic outcomes.

## 1. Introduction

Neoadjuvant chemotherapy (NACT) is used with increasing frequency in the multidisciplinary management of locally advanced breast cancer (LABC). Patients that will most probably benefit from NACT include high tumor-to-breast volume ratio; ref. [1] lymph-node-positive [2] and specific biological features of primary cancer as high-grade disease; [3] hormone receptor negative breast cancer (BC); triple negative (TN) [4] and overexpression of human epidermal growth factor receptor 2 (HER2+) phenotypes, even at an early stage (EBC).

A key benefit of NACT is to downstage the tumor to favor breast-conservative surgery (BCS) over conservative mastectomy with reconstruction (CMR) [5,6,7], avoid axillary dissection with its related morbidity if there was no evidence of lymph node involvement at sentinel lymph node biopsy [8,9], and the evaluation of an in vivo response to NACT that could guide the prescription of further personalized adjuvant chemotherapy [10,11,12,13,14,15,16,17]. The two main goals of the surgeon performing BCS are to obtain tumor-free excision margins preserving the healthy tissue and an adequate aesthetic outcome. Tumor-involved margins must be avoided because this condition increases the risk of local recurrence.

In order to optimize oncological and aesthetic outcomes in patients with large or multifocal tumors desiring breast conservation, oncoplastic surgery (OPS) can be used after NACT [18]. The indication for OPS is a nonoptimal response after NAC, for which a BCS with safe margins would either seem impossible or lead to a major deformity [19]. CMR remains indicated in patients with multicentric disease, tumor volume to breast ratio that requires the excision of more than 50% of the glandular tissue, widespread microcalcifications, and pathogenic variants of BRCA 1/2 genes (BRCA) [20,21,22].

The choice of the surgical technique is usually based on tumor characteristics (size and location), extent of resection, breast features (volume, shape, and glandular density), previous surgery, and patient expectations or wishes. However, the selection of optimal surgery after NACT is often difficult and based only on surgeons’ expertise, especially for cancers that did not shrink optimally when large glandular resections were required. 

The aim of the study was to create a standardized scoring system named pNESSy (post-NEoadjuvant Score System) that can indicate the most appropriate surgical treatment in order to optimize oncological and aesthetic outcomes in breast cancer patients after NACT. The secondary aim was to evaluate the ability of the pNESSy to guarantee an adequate outcome defined as locoregional disease free-survival (LR-DFS), distant disease-free survival (DDFS), and overall survival (OS) and aesthetic result.

## 2. Materials and Methods

This was a monocentric, retrospective study conducted at Fondazione Policlinico Universitario A. Gemelli IRCCS, in Rome, Italy. Figure 1 shows the consort diagram with study characteristics.

This study evaluated patients with breast cancer (stage I–III) who underwent NACT and subsequent surgery according to international guidelines [23] between January 2016 and March 2021. The inclusion and exclusion criteria are reported in Figure 1.

Data collection from patient records was prospectively updated in a database. This study was conducted in accordance with the ethical standards as laid down in the Declaration of Helsinki and registered at ClinicalTrials.gov (NCT05213403) [24]. The protocol was approved by Central Ethics Committee (number: RS 4694).

The indication to NACT was decided by a multidisciplinary team (MDT) composed of breast and plastic surgeons, oncologists, radiotherapists, radiologists, pathologists, psychologists, geneticists, and a case manager. Diagnostics and therapeutic management are shown in Figure 2.

Subsequently, patients were evaluated to obtain locoregional and systemic staging [25]. Baseline assessment included outpatient’s evaluation, histological definition and locoregional and systemic staging. Magnetic resonance imaging (MRI) and RX-mammography were revised to define cancer extension, focality, prediction of the excised breast area (PEBA), evaluated with pre-NACT MRI (MRI-PEBA) and RX-mammography (RX-PEBA) (Appendix A), Rancati score [26], and microcalcification extension. Information on grade of ptosis was collected [27] (Appendix A).

Neoadjuvant schedules were decided in accordance with international guidelines [23].

Preoperative restaging was performed using clinical re-evaluation, breast and axillary ultrasound, RX-mammography, and MRI [28]. Clinical response was assessed according to RECIST1.1 criteria [29]. Post-NACT images were reviewed to evaluate tumor extension (MRI largest diameter), residual disease, focality, post-NACT MRI-PEBA, and RX-PEBA (Figure 2).

### 2.1. Operative Protocol, Surgical Technique, and Pathological Evaluation

Patients included in the study underwent three types of surgery: BCS, OPS, and CMR. Surgical planning was always discussed in a multidisciplinary meeting. Indication for OPS was poor response after NACT for which a BCS with safe margins would either seem not possible or lead to deformity, high tumor volume to breast ratio, or multifocal cancer. CMR was indicated in patients with extensive or multicentric cancers and tumor volume to breast ratio that required the excision of more than 50% of tissue volume, inability to obtain clear surgical margins with OPS, contraindications to adjuvant radiotherapy, and patient preference. Bilateral CMR was performed in patients with bilateral breast tumor or in women with unilateral disease and a high risk of contralateral cancer, such as BRCA mutation carriers.

OPS included “inverted T mammoplasty”, “J mammoplasty”, “round block technique”, and “batwing mammoplasty”. CMR included nipple-sparing and skin-sparing mastectomy with breast prosthetic reconstruction. 

During surgery, all patients underwent additional surgical cavity shavings to evaluate any neoplastic infiltration adjacent to the surgical specimen [30]. The axillary surgical approach was based on the clinical response to NACT. Patients with post NACT clinically positive nodes (ycN+) directly underwent axillary dissection (AD). Patients with post-NACT clinically negative nodes (ycN0) underwent sentinel lymph-node biopsy (SLNB), and AD was performed in case of metastases in sentinel nodes.

Pathological evaluation consists of 1. surgical specimen volume considered as an ellipsoid [31]; 2. margin involvement by either invasive cancer (IC) or ductal carcinoma in situ (DCIS); 3. presence of either IC or DCIS on cavity shavings. The failure of “oncological radicality” was defined as the presence of “ink on tumor” for both IC and DCIS or distance less than 2 mm for DCIS in the specimen, or as the presence of any kind of tumor in the cavity shavings [32,33,34].

### 2.2. Adjuvant Treatment

Adjuvant therapy was assessed by the MDT in relation to pre-NACT staging, type of surgery, tumor biology, and pathological staging. In case of TN or HER2 + tumor with residual disease, patients received capecitabine or Trastuzumab-Emtansine (TDM-1), respectively, according to active guidelines [4,35,36,37]. ER/PGR-positive patients received adjuvant hormonal treatment according to menopausal status. Radiation therapy was advised in accordance with international guidelines [38].

### 2.3. Statistical Analysis and Score Processing

Statistical analysis was performed using SPSS (Statistical Package of Social Science), version n.27. Continuous variables were described by mean ± standard deviation (median; interquartile range) and compared with ANOVA; categorical variables were summarized by absolute number and percentage and compared using chi-square test. For pNESSy, we evaluated factors potentially associated with the type of surgery: age at diagnosis, BMI, pathogenic mutations of BRCA, bra size, Rancati score, microcalcifications extension, ptosis, breast volume measured at pre-NACT MRI; multifocality/multicentricity, pre-NACT RX-PEBA and MRI-PEBA; post-NACT radiological response, multifocality/multicentricity, post-NACT RX-PEBA and MRI-PEBA. For categorical variables, the association was assessed using chi-square test and confirmed by univariate analysis. Continuous variables were divided into 10 identical groups (deciles), and the association of each decile with the type of surgery was evaluated using logistic regression. Deciles results related with the same surgery were considered in a single interval. Each interval was associated with the type of surgery and confirmed with logistic regression. Intervals significantly associated with a type of surgery at univariate analysis were evaluated in the multivariable analysis until the best association between factors was identified. The factors linked with the same surgery turned out to be the entries that composed pNESSy. The value of each single interval was the coefficient β at the multivariable analysis. The procedure was internally validated using a bootstrap technique based on 1000 samples. The diagnostic performance of our model was evaluated using area under the curve (AUC) of the receiver operating characteristics curve (ROC). Survival curves were obtained with the Kaplan–Meier method and compared using log-rank test. Statistical evaluations were performed considering two-tailed models, and differences were considered significant if *p*-value < 0.05. In the model definition we accepted significant values until *p* < 0.10.

### 2.4. Evaluation of Oncological, Aesthetic Outcomes, and Patient Quality of Life (QoL)

Patients were evaluated every 6 months by outpatient visits, including clinical examination, execution of blood chemistry tests with essay of tumor markers, breast US and mammography every 6 months, and systemic staging by TC-TB or PET-TC scan every year.

To assess oncological outcomes, we used LR-DFS, defined as months between start of NACT and date of neoplastic recurrence in the ipsilateral residual mammary gland, chest wall, or axilla; D-DFS, months between start of NACT and date of onset of visceral or skeletal metastases; and OS, months between start of NACT and death, or censored at the date of last follow-up.

The evaluation of aesthetic outcomes and patient quality of life (QoL) was obtained by administering the following BREAST-Q^©^ questionnaires: satisfaction with breasts (questionnaire 1); psychological wellbeing (questionnaire 2); sexual wellbeing (questionnaire 3); and physical wellbeing: chest (questionnaire 4). Finally, we evaluated residual breast sensitivity and its influence on daily life with a questionnaire shower in Appendix A.

Answers of all questionnaires were divided into three groups: 1. poor outcome (between 0–40); 2. acceptable outcome (41–70); 3. excellent outcome (71–100).

### 2.5. Assessment of the Adherence of Score with Type of Surgery

Patients were divided into two groups according to correspondence (G1) or discrepancy (G2) among score and type of performed surgery; oncological, aesthetics outcomes, and patient QoL were subsequently analyzed in the two groups.

## 3. Results

A total of 989 patients underwent NACT. We excluded 693 patients (70.1%) due to unavailability of imaging, 2 (0.2%) with synchronous cancer, 8 (0.8%) with previous breast cancer, 9 (0.9%) with evidence of widespread disease, and 20 patients who underwent MRM. The remaining 255 patients (25.8%) were included in the analysis: 118 (46.3%) underwent BCS, 49 (19.2%) OPS, and 88 (34.5%) CMR.

### 3.1. Demographic, Clinical, and Biological Features

Table 1 reports demographic, clinical, and biological features of patients enrolled.

Mean age was 49.1 ± 10.8 y and BMI 25 ± 4.7 kg/m^2^. Mean values differed among three group: BCS group had a mean age of 51.9 y and a BMI of 25.4 kg/m^2^, OPS group, respectively, of 48.3 y and 27.4 kg/m^2^, and CMR group of 45.8 y and 23.2 kg/m^2^ (Appendix A).

Patients undergoing BCS had a size 3 (42.4%) and grade 1 ptosis (44.9%) more frequently than patients undergoing OPS (size 4 or greater: 57.2% and ptosis grade 2 or 3: 63.3%) and CMR (size: <2–44.4% and ptosis grade: 0–50%). BRCA mutation was more common in CMR. No difference was found in breast quadrant involved, cigarette smoking, comorbidities, and tumor biological features.

### 3.2. Pre- and Post-NACT Radiological Assessment

Table 2 shows the radiological assessment before and after NACT. 

The groups differed in breast volume (CMR < 645.99 cm^3^; OPS > 1009.40 cm^3^ and BCS with intermediate value); disease focality; pre-NACT RX-PEBA (BCS < 0.44; OPS 0.45–1.35; CMR > 1.36) and pre-NACT MRI-PEBA (BCS < 3.52; OPS 3.53–9.99; CMR > 10). Differences were found in number of involved quadrants, Rancati score, and microcalcification extension (Appendix A). No difference was found in clinical T and N stage. At post-NACT radiological assessment, disease focality, RX-PEBA (BCS < 0.041; OPS 0.042–4.61; CMR > 4.62) and MRI-PEBA (BCS < 0.26; OPS 0.27–1.29; CMR > 1.30) differed between the three groups (Table 2). Radiological complete response (rCR) was more common in the BCS group. No other differences were found among groups.

### 3.3. Pathological Response

Overall, 139 (54.5%) patients achieved a pCR. Among these, 19 showed the presence of DCIS. Seventy-one patients (27.8%) achieved a pCR on nodes, 41 (16.1%) simultaneously achieved a pCR on breast and nodes (Appendix A). No differences were found regarding residual tumor histotype, hormone receptors, and clinical prognostic factors. A significant difference was observed in surgical specimen volume (which was progressively larger from BCS to CMR) (*p* < 0.0001) and in presence of DCIS, which was more evident in OPS (51.3%) compared to BCS and CMR (28% and 35.2%) (*p* = 0.009).

### 3.4. Adjuvant Treatments

All patients undergoing BCS and OPS received radiation therapy on residual mammary gland. Fifty-seven patients (64.8%) undergoing CMR received chest wall radiation therapy. Radiation therapy of the supra/subclavicular lymph node was performed in 147 patients (57.6%). There was no statistical difference among the three groups (*p* = 0.432).

One hundred and sixty-nine patients (66.3%) received hormone therapy, 4 (1.6%) with Tamoxifen alone (TAM), 23 (9%) with TAM and LH-RH, 93 (36.5%) with aromatase inhibitors (AI), and 49 (19.2%) with AI and LH-RH. Patients undergoing CMR were more frequently subjected to the use of AI plus LH-RH (*p* = 0.013).

Seventeen (6.7%) HER2-positive patients without pCR performed T-DM1 as standard of care. Seventeen (6.7%) TN patients without pCR received capecitabine. No statistically significant differences were found between groups concerning adjuvant treatment (*p* = 0.960). One hundred and sixty-two patients (63.5%) did not receive adjuvant chemotherapy.

### 3.5. Outcomes According to the Type of Surgery

No differences were observed for the surgical techniques (LR-DFS, DDFS, and OS).

### 3.6. Definition of “pNESSy”

For the development of the score, we initially defined, by univariate and multivariable analysis, the anatomical, pre-, and post-NACT radiological features associated with each type of intervention. Subsequently, the best association between significative features were found and used for scoring definitions. 

#### 3.6.1. Univariate and Multivariable Analysis for BCS (Appendix A)

The multivariate analysis resulted in six independent variables: ptosis 1 (*p* = 0.098); breast volume between 646.00 and 1009.39 measured with MRI (*p* = 0.012); unifocal neoplasm at diagnosis (*p* < 0.0001); pre-NACT MRI-PEBA lower than 3.52 (*p* = 0.001); post-NACT complete radiological response or unifocality (*p* = 0.001); and post-NACT RX-PEBA <0.041 (*p* = 0.001).

#### 3.6.2. Univariate and Multivariable Analysis for OPS (Appendix A)

In multivariate analysis, six variables were found to be independently associated with OPS: grade 2 or 3 of ptosis (*p* = 0.002), breast volume greater than 1009.4 (*p* = 0.002); multifocality involving one or two quadrants adjacent at diagnosis (*p* = 0.012); pre-NACT MRI-PEBA between 3.53 and 9.99 (*p* = 0.006); post-NACT multifocality (*p* = 0.001); and finally post-NACT RX-PEBA between 0.27 and 1.29 (*p* = 0.001).

#### 3.6.3. Univariate and Multivariable Analysis for CMR (Appendix A)

Seven variables were significant in CMR: BRCA mutation (*p* = 0.001); breast ptosis 0 (*p* = 0.009); breast volume less than 645.99 (*p* = 0.004); multicentric lesions (*p* = 0.001); pre-NACT MRI-PEBA greater than 10 (*p* = 0.001); post-NACT multicentricity (*p* = 0.005); and post-NACT RX-PEBA greater than 4.62 (*p* = 0.020).

At multivariable analysis, the best associations in the score were BRCA mutation, breast ptosis, breast volume, pre- and post-NACT disease focality, pre-NACT MRI-PEBA, and post-NACT RX-PEBA. The value of coefficient β was associated with each variable. The sum of the seven coefficients resulted in the score (Table 3).

A score between 6.896 and 8.724 was associated with BCS, between 8.725 and 9.375 with OPS, and between 9.376 and 14.245 with CMR. AUC was, respectively, 0.835, 0.766, and 0.825 (Figure 3).

### 3.7. Assessment of pNESSy Adherence with Surgery and Evaluation of Outcomes

One hundred and sixty patients (62.7%) showed a correspondence between score and surgery (G1), while 95 cases (37.3%) showed a discrepancy (G2) (Figure 4). 

Between the two groups, we found no differences regarding radiotherapy (in G1, 146–88.5% of patients underwent RT, in G2, 78–86.7% (*p* = 0.691)) and second surgery (one patient in G1 and one patient in G2 (*p* = 0.669)).

Two hundred and thirty-three patients (91.4%) achieved “oncological radicality” with tumor-free specimen margins and cavity shavings. Twenty-two cases did not obtain radicality: 15 patients had “ink on invasive tumor”, 4 patients had “ink on in situ tumor”, and 3 patients had evidence of cancer on cavity shavings. Oncological radicality was more frequently achieved in G1 patients (95% versus 85.3%; *p* = 0.010) (Table 4).

After a follow-up of 40.5 ± 16 months (39.6; 27–51.5), eight patients had locoregional relapses (seven in breast and one in axilla). Three breast relapses and the axillary recurrence were observed in the BCS group, one breast relapse in the OPS group, and three in the CMR group (overall LR-DFS of 96.4%, 95.5%, and 93%, respectively; *p* = 0.928) (Figure 5A). When stratified for adherence with pNESSy, LR-DFS was 98.8% for G1 and 88.9% for G2 *(p* < 0.001) (Figure 6A).

We observed 29 distant relapses: 12 (4.7%) in the BCS group, 5 (1.9%) in the OPS group, and 12 (4.7%) in the CMR group. Difference in overall DDFS regarding type of surgery did not reach significance (respectively, 65.1%, 87.4%, and 81.6%; *p* = 0.723) (Figure 5B). Furthermore, DDFS was not different between G1 and G2 (79.1% and 80.7%; *p* = 0.200) (Figure 6B).

We observed 15 deaths due to systemic progression of disease: 6 (2.4%) in the BCS group, 3 (1.2%) in the OPS group, and 6 (2.4%) in the CMR group. There was no significant difference in OS between type of surgery (92.2%, 93.7%, and 91.6%, respectively; *p* = 0.892) (Figure 5C). The analysis of adherence to pNESSy instead showed a lower incidence of events in the G1 group compared to G2 (96.1% vs. 86.5%; *p* = 0.017) (Figure 6C).

Data concerning aesthetical outcomes were collected in 195 patients (76.5%) who submitted a reply to the administered questionnaires. G1 patients reported both a better satisfaction with breasts (39.8% vs. 27.5% among the very satisfied; *p* = 0.017) and better physical wellbeing (93.5% vs. 73.6%; *p* = 0.001).

## 4. Discussion

NACT is being used with increasing frequency in the multidisciplinary management of patients with breast cancer [13,14,16,19,33,34,35,36,37,38,39,40,41].

Thanks to downstaging of tumor obtained with NACT, the indications for BCS can also be expanded in LABC that initially are candidates for mastectomy [42]. An appropriate conservative surgical treatment should always ensure the achievement of two goals: oncological radicality and optimal aesthetic outcomes [43,44,45]; oncological radicality means obtaining tumor-free margins and minimizing the risk of local recurrence, while optimal aesthetic outcomes require the preservation of an adequate and harmonious shape of the breast, which should always be symmetrical to the contralateral one, removing as little as possible [46,47,48].

Achieving both goals with standard BCS is not always easy, and oncoplastic techniques allow the resection of more breast tissue with safe margins and appropriate cosmetic results, often avoiding the need for mastectomy in patients with a partial response [49].

However, the selection of the more appropriate surgery after NACT is often difficult, especially for tumors that did not shrink optimally when large glandular resections were required; the limits of preoperative imaging in accurately defining the extent of the residual tumor, the persistence of DCIS foci, and the phenomenon of tumor fragmentation, are all factors that can make the surgical choice more complex [50].

To date, no specific tool supports the surgeon in choosing surgery in neoadjuvant settings; the selection of the surgical technique is usually based on tumor characteristics (histotype, size, and location), extent of resection, breast features (volume, shape, and glandular density), previous surgery, wishes of the patient, and surgeons’ expertise; the presence of multifocality, extensive microcalcifications, and a lobular histotype were described as predictive factors for mastectomy [51,52,53,54].

No author generated a predictive score system to select the optimal surgical procedure (BCS, OPS, and CMR) based on pre- and post-NACT characteristics; therefore, the aim of our study was to design a standardized scoring system that can indicate the most appropriate surgical treatment to optimize oncological and aesthetic outcomes in breast cancer patients after NACT.

We evaluated genetic, anatomic, and radiologic features as BRCA mutations, breast ptosis, breast volume, pre- and post-NACT multifocality and multicentricity, pre-NACT MRI-PEBA, and post-NACT RX-PEBA. The choice of RX-PEBA post-NACT with respect to the presence of microcalcifications alone in the evaluation of the surgical procedure allowed the evaluation of the persistence of opacities without microcalcifications. In the same way, pre-NACT MRI-PEBA allows us to take into account distant foci that are not evaluable on mammograms, or that resolve completely after NACT. Finally, we identified seven values that should be considered to determine the final score and guide the surgical choice: 1. Presence of mutation of BRCA; 2. degree of breast ptosis; 3. breast volume evaluated with MRI; 4. presence of foci/centricity pre-NACT; MRI-PEBA pre-NACT; 5. presence of foci/centricity post-NACT; 6. RX-PEBA post-NACT.

The score obtained, named pNESSy, showed values between 6.896 and 14.245. A score between 6.896 and 8.724 was predictive for BCS, between 8.725 and 9.375 for OPS, and between 9.376 and 14.245 for CMR. 

In our series, patients undergoing the three types of surgery showed similar characteristics in both biological features (immunophenotype, histotype, and grading) and stage at diagnosis (*p* = 0.904). The only difference between the three groups relates to the presence of a BRCA mutation, a factor associated more frequently with CMR [55]. Six BRCA mutated patients did not undergo CMR, either because the mutation was identified after surgery or because of patient preference.

In accordance with the literature, there was no difference in terms of oncological safety of the three surgical procedures [47,56,57,58].

Subsequently, in order to verify pNESSy, all patients were divided into two groups according to correspondence (G1) or discrepancy (G2) between score and type of surgery performed; oncological, aesthetics outcomes, and patient QoL were subsequently analyzed in the two groups.

With regard to oncological safety, the results showed that pNESSy could facilitate the achievement of tumor-free margins and minimize the risk of local recurrence. After a median follow-up of 40 months, G1 compared to G2 patients showed a lower rate of involved margins (8 versus 14; *p* = 0.010), a better LR-DFS (98.8 versus 88.9; *p* < 0.001), and a better OS (96.1% versus 86.5%; *p* = 0.017). Regarding the features associated with oncological aggressiveness (TN, HER2+, and initial axillary involvement), no difference was found between the two groups, and the difference in terms of LR-DFS and OS cannot be attributed to these factors. Possible reasons for worse OS in the G2 group could be a lower incidence of pCR in breast (*p* = 0.027) and axilla (*p* = 0.005) [59], a greater number of cT3/4 at diagnosis (*p* = 0.046), and a higher incidence of locoregional recurrences *(p* = 0.001) [60].

With regard to aesthetic outcomes and patient QoL, the results showed that pNESSy could help surgeons and patients to choose the best performing type of surgery between BCS, OPS, and CMR without compromising oncological safety. The assessment of BREAST-Q^©^ questionnaires indicates that G1 patients had better outcomes in terms of satisfaction with breasts and physical wellbeing compared to G2 patients. A possible explanation for this difference could be that this score helps the surgeon to avoid overtreatment by easing the preservation of breast shape and volume. The use of OPS techniques can allow preservation of the body image and the breast conservation when standard BCS would result in major deformities [61].

This study presents several strengths. “pNESSy” constitutes the first standardized tool to determine the optimal surgery after NACT by taking into account clinical, radiological, and genetic parameters. Moreover, no author described a model able not only to minimize recurrences, but also to optimize aesthetic outcomes. Patient satisfaction and QoL were included in our model by the use of a standardized tool, the BREAST-Q^©^ questionnaire. In addition, choice of procedure is guided by factors related to both initial extension and response to therapy. However, the pNESSy presents also some limitations because it was created thanks to a retrospective analysis of a single institution with a limited number of patients and a short follow-up.

## 5. Conclusions

An innovative scoring system based on clinical and radiological characteristics was created to select the most appropriate surgical treatment for breast cancer patients after NACT; the use of this tool allows us to reduce the risk of local recurrence and optimize the aesthetic results by improving the patient’s QoL, especially for tumors that do not shrink optimally; however, further, high-quality multicenter trials are needed to definitively validate our scoring system and overcome the aforementioned limitations.

## Figures and Tables

**Figure 1 jpm-13-01280-f001:**
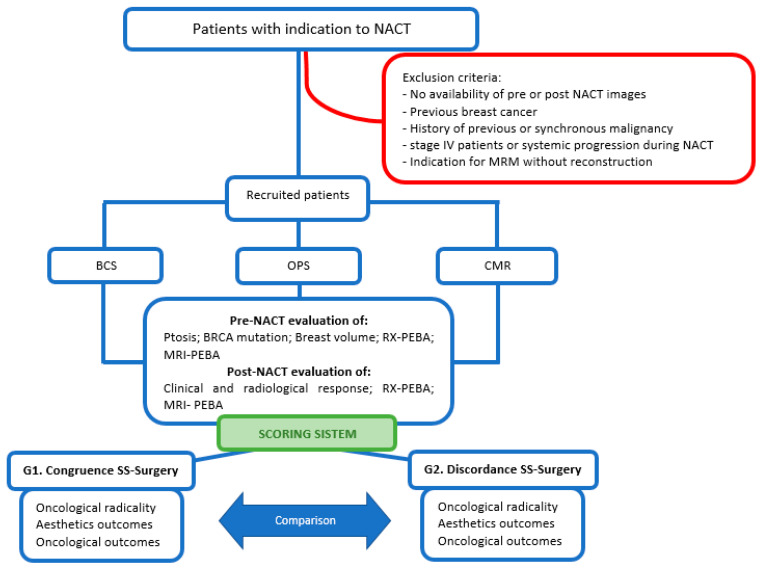
Consort diagram. Patients with breast cancer and indication to NACT and subsequent surgery were divided into three groups: breast-conserving surgery (BCS), level II oncoplastic surgery (OPS), and conservative mastectomy with reconstruction (CMR). An evaluation of the pre- and post-NACT clinical and radiological characteristics was performed with the aim of defining a scoring system that could predict the best surgery. Finally, two groups on the basis of the correspondence between score and surgery were defined and compared on the basis of “oncological radicality”, aesthetic, and oncological outcomes.

**Figure 2 jpm-13-01280-f002:**
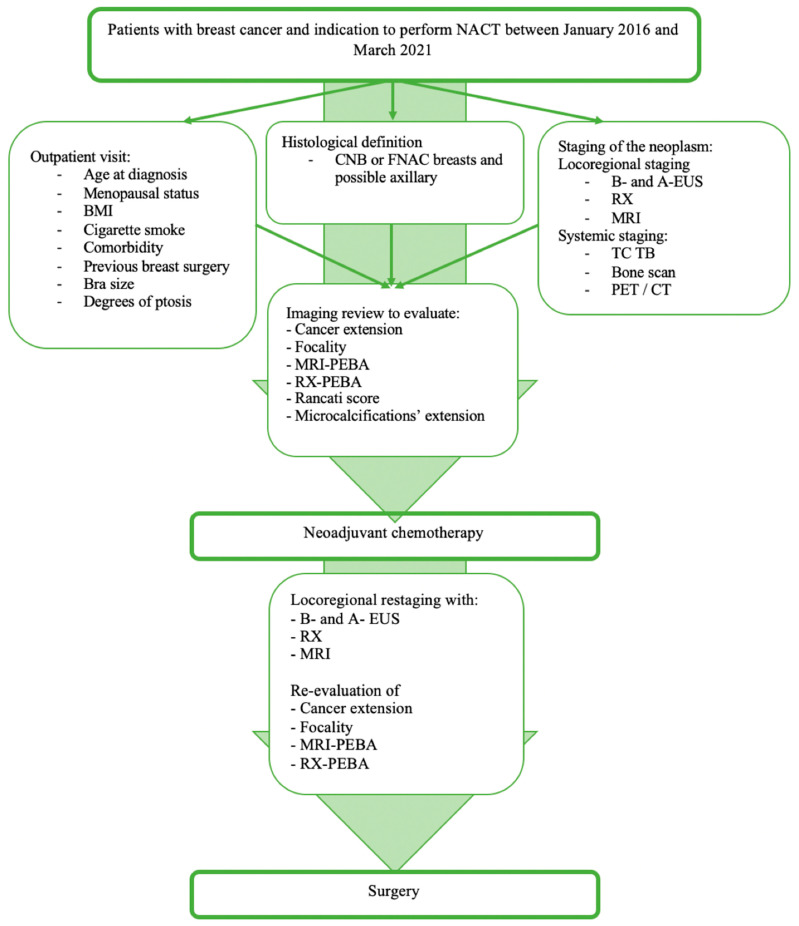
Diagnostic–therapeutic flow-chart of the patients included in the study. Imaging revision was performed immediately prior to the analysis for the formulation of the score. Surgery was evaluated on a multidisciplinary basis. BMI: body mass index; EUS: ecographic ultrasound; RX: RX-mammography; CNB: core needle biopsy; FNAC: fine needle aspiration cytology; RX-PEBA: prevision of excised breast area evaluated with mammography; MRI-PEBA: prevision of excised breast area evaluated with magnetic resonance imaging (eFigure1).

**Figure 3 jpm-13-01280-f003:**
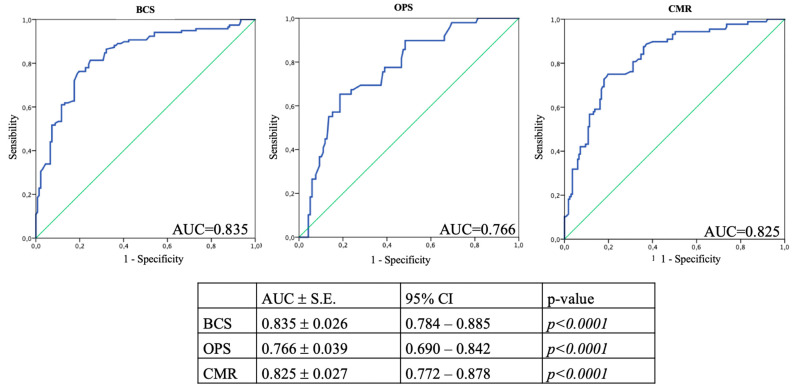
Description of the ROC curves associated with the type of surgery. The area under the curve (AUC) relating to OPS is calculated by excluding patients undergoing CMR; therefore, on a total of 167 patients. The lower AUC value shown for OPS could be explained by the reduced number of patients undergoing this type of surgery. BCS: breast-conservative surgery; OPS: level II oncoplastic surgery; CMR: conservative surgery with reconstruction.

**Figure 4 jpm-13-01280-f004:**
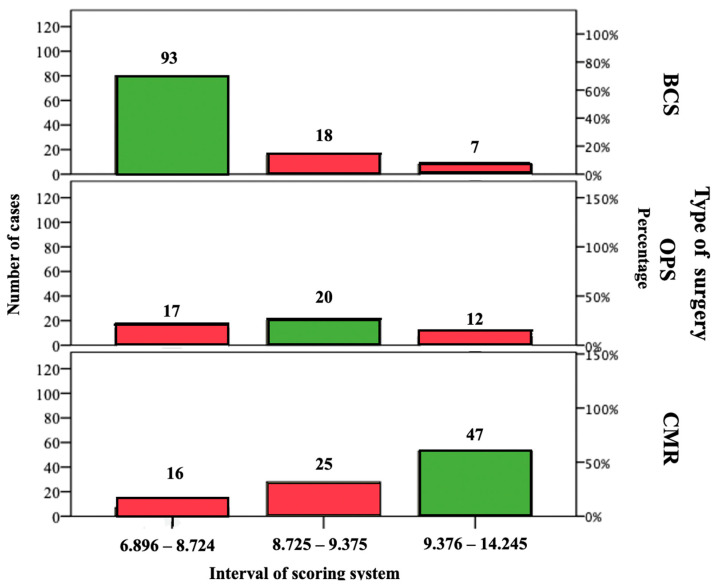
Adherence between pNESSy and surgery performed. One hundred and sixty patients underwent surgery congruent with the score: ninety-three were subjected to BCS, twenty to OPS, and forty-seven to CMR (G1 group). The remaining patients (95–37.2%) received an operation that was not congruent with pNESSy (G2 group).

**Figure 5 jpm-13-01280-f005:**
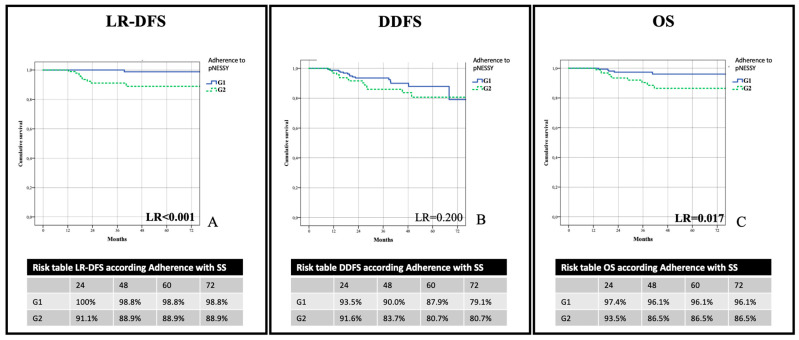
Evaluation of oncological outcomes between groups according to adherence to pNESSy. Locoregional recurrence-free survival (LR-DFS) (**A**), distant disease-free survival (DDFS) (**B**), and Overall survival (OS) (**C**) according to adherence with surgery and pNESSy. G1 patients with adherence among pNESSy and surgery; G2: patients with difference between pNESSy and surgery.

**Figure 6 jpm-13-01280-f006:**
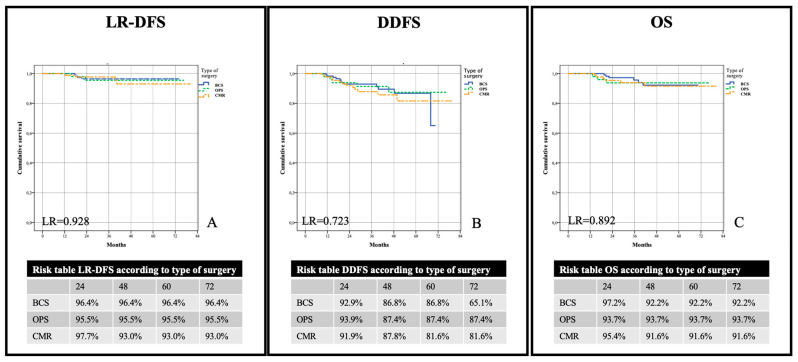
Evaluation of oncological outcomes according to type of surgery. Locoregional recurrence-free survival (LR-DFS) (**A**), distant disease-free survival (DDFS) (**B**), and overall survival (OS) (**C**) according to type of surgery.

**Table 1 jpm-13-01280-t001:** Demographic, anatomic–morphological mammary, and biological characteristics of the enrolled patients.

	BCS 118 (46.3%)	OPS 49 (19.2%)	CMR 88 (34.5%)	*p*-Value
Demographic characteristics				
Age (y)	51.9 ± 11.5 (44.2–60.3)	48.3 ± 9.1 (42.2–53)	45. 8 ± 9.7 (39.6–52.4)	*p* < 0.0001
BMI (kg/m^2^)	25.4 ± 4.7 (22.7–27.3)	27.4 ± 5.3 (23.3–30.1)	23.2 ± 3.5 (20.8–24.5)	*p* < 0.0001
Menopausal status - Yes - No	58 (49.2%) 60 (50.8%9	19 (38.8%) 30 (61.2%)	24 (27.3%) 64 (72.7%)	*p* = 0.006
Pathological mutations of BRCA 1 or 2	5 (4.2%)	1 (2%)	18 (20.5%)	*p* < 0.0001
Anatomic-morphological mammary characteristics of breasts				
Bra size - 1–2 - 3 - >4	26 (22.0%) 50 (42.4%) 42 (35.6%)	8 (16.3%) 13 (26.5%) 28 (57.2%)	8 (44.4%) 30 (34.1%) 19 (21.5%)	*p* < 0.0001
Ptosis - Grade 0 - Grade 1 - Grade 2 or 3	20 (16.9%) 53 (44.9%) 45 (38.2%)	5 (10.2%) 13 (26.5%) 31 (63.3%)	44 (50%) 30 (34.1%) 14 (15.9%)	*p* < 0.0001
Location of cancer				
- Upper-outer Q - Upper-internal Q - Infer-internal Q - Infer-outer Q - Retro-areolar Q	78 (66.1%) 8 (6.8%) 4 (3.4%) 20 (16.9%) 8 (6.8%)	36 (73.5%) 4 (8.2%) 1 (2%) 6 (12.2%) 2 (4.1%)	65 (73.9%) 7 (8%) 2 (2.3%) 11 (12.5%) 3 (3.4%)	*p* = 0.948
Biological characteristics of cancer				
Histotype - DIC - LIC - IC NST *	86 (72.9%) 8 (6.8%) 24 (20.3%)	35 (71.5%) 3 (6.1%) 11 (22.4%)	63 (71.6%) 6 (6.8%) 19 (21.6%)	*p* = 0.997
Grading - 1 - 2 - 3	0 (0%) 33 (28%) 85 (72%)	0 (0%) 15 (30.6%) 34 (69.4%)	1 (1.1%) 24 (27.3%) 63 (71.6%)	*p* = 0.830
Immunophenotype - Luminal/HER2 − - HER2 + - TN	59 (50%) 39 (33.1%) 20 (16.9%)	22 (44.9%) 16 (32.7%) 11 (22.4%)	38 (43.2%) 31 (35.2%) 19 (21.6%)	*p* = 0.825
Initial stage of cancer **				
- I - II A - II B - III A - III B - III C	6 (5.1%) 32 (27.1%) 40 (33.9%) 29 (24.6%) 7 (5.9%) 4 (3.4%)	1 (8.3%) 9 (18.4%) 18 (36.7%) 18 (36.7%) 2 (4.1%) 1 (20%)	5 (5.7%) 24 (27.3%) 26 (29.5%) 24 (27.3%) 6 (6.8%) 3 (3.4%)	*p* = 0.904

* invasive carcinoma no special type; ** AIOM Guidelines 2020.

**Table 2 jpm-13-01280-t002:** Radiological assessment pre- and post-neoadjuvant treatment.

	BCS 118 (46.3%)	OPS 49 (19.2%)	CMR 88 (34.5%)	*p*-Value
Assessment Pre-NACT				
Rancati score - 1 - 2 - 3	31 (26.3%) 64 (54.2%) 23 (19.5%)	8 (16.3%) 30 (61.3%) 11 (22.4%)	39 (44.3%) 43 (48.9%) 6 (6.8%)	*p* = 0.001
Breast volume (cm^3^) * - <645.99 - 646.00–1009.39 - >1009.40	33 (28%) 53 (44.9%) 32 (27.1%)	5 (10.2%) 16 (32.7%) 28 (57.1%)	53 (60.2%) 24 (27.3%) 11 (12.5%)	*p* < 0.0001
Microcalcification extension - <21.9 mm - 22–79.9 mm - >80 mm	90 (76.3%) 27 (22.9%) 1 (0.8%)	20 (40.8%) 26 (53.1%) 3 (6.1%)	42 (47.7%) 36 (40.9%) 10 (10.4%)	*p* < 0.0001
No. involved quadrants - 1 - 2 - >3	77 (65.3%) 31 (26.3%) 10 (8.5%)	17 (34.7%) 16 (32.7%) 16 (32.7%)	29 (33.0%) 18 (20.5%) 41 (46.6%)	*p* < 0.0001
Pre-NACT RX-PEBA - <0.44 - 0.45–1.35 - >1.36	71 (60.2%) 33 (28.1%) 14 (11.7%)	14 (28.6%) 30 (61.2%) 5 (10.2%)	20 (22.7%) 12 (13.6%) 56 (63.7%)	*p* < 0.0001
Pre-NACT focality - Unifocality - Multifocality - Multicentricity	84 (71.2%) 27 (22.9%) 7 (5.9%)	12 (24.5%) 34 (69.4%) 3 (6.1%)	28 (31.8%) 17 (19.3%) 43 (48.9%)	*p* < 0.0001
Pre-NACT MRI-PEBA - <3.52 - 3.53–9.99 - >10.00	74 (62.7%) 38 (32.2%) 6 (5.1%)	10 (20.4%) 33 (67.4%) 6 (12.2%)	19 (21.6%) 30 (34.1%) 39 (44.3%)	*p* < 0.0001
Assessment post-NACT				
Radiological response - Complete - Partial - Stable - Progression	66 (55.9%) 38 (32.3%) 12 (10.2%) 2 (1.7%)	16 (32.7%) 29 (59.2%) 4 (8.2%) 0 (0%)	38 (43.2%) 33 (37.5%) 16 (18.2%) 1 (1.1%)	*p* = 0.016
Post-NACT RX-PEBA - <0.041 - 0.042–4.61 - >4.62	87 (73.7%) 27 (22.9%) 4 (3.4%)	14 (28.6%) 35 (71.4%) 0 (0%)	27 (30.7%) 40 (45.5%) 21 (9.8%)	*p* < 0.0001
Post-NACT focality - Complete response - Unifocality - Multifocality - Multicentricity	65 (55.1%) 42 (35.6%) 9 (7.6%) 2 (1.7%)	15 (30.6%) 12 (23.9%) 20 (40.8%) 2 (4.1%)	36 (40.9%) 21 (23.9%) 6 (6.8%) 25 (28.4%)	*p* < 0.0001
Post-NACT MRI-PEBA - <0.26 - 0.27–1.29 - >1.30	75 (63.6%) 36 (30.5%) 7 (5.9%)	14 (28.6%) 24 (49%) 11 (22.4%)	39 (44.3%) 16 (18.2%) 33 (37.5%)	*p* < 0.0001

* Evaluated with MRI; PEBA = prediction of excised breast area. BCS: breast-conserving surgery; OPS: level II oncoplastic surgery; CMR: conservative mastectomy with reconstruction.

**Table 3 jpm-13-01280-t003:** Scoring system (pNESSy) for surgery after NACT.

BRCA 1 or 2 Genes	No Pathogenetic Variant		Pathological Mutation
	0		2.867
Ptosis	Grade 0	Grade 1	Grade 2 or 3
	1.389	0.511	1.352
Breast volume evaluated with MRI	<645.99 cm^3^	646.00–1009.39 cm^3^	>1009.4 cm^3^
	1.375	0.900	1.526
Pre-NACT Focality	Unifocality	Multifocality	Multicentricity
	1.368	1.193	2.309
Pre-NACT MRI-PEBA	<3.52	3.53–9.99	>10.00
	1.251	1.391	1.860
Post-NACT Focality	Clinical complete response or unifocal	Multifocality	Multicentricity
	1.536	2.274	2.007
Post-NACT RX-PEBA	<0.041	0.042–4.61	>4.62
	1.505	2.020	1.589
SCORE	**6.896–8.724**	**8.725–9.375**	**9.376–14.245**
	***p* < 0.0001**	***p* < 0.0001**	***p* < 0.0001**
TYPE OF SURGERY	**BCS**	**OPS**	**CMR**

PEBA = prediction of excised breast area. BCS: breast-conserving surgery; OPS: level II oncoplastic surgery; CMR: conservative mastectomy with reconstruction.

**Table 4 jpm-13-01280-t004:** Evaluation of oncological radicality and aesthetic outcomes based on adherence to pNESSy.

	G1 164 (64.3%)	G2 91 (35.7%)	*p*-Value
Oncological radicality			
Reached Failed	152 (95%) 8 (5%)	81 (85.3%) 14 (14.7%)	*p* = 0.010
Aesthetic Outcomes evaluate on patients who responded to “Breast Q” questionnaire *			
	G1 123 (75%)	G2 69 (75.8%)	
Q1. Satisfaction with breasts <40 41–70 >71	29 (23.6%) 45 (36.6%) 49 (39.8%)	30 (43.5%) 20 (29.0%) 19 (27.5%)	*p* = 0.017
Q2. Psychological wellbeing <40 41–70 >71	21 (17.1%) 37 (30.1%) 65 (52.8%)	18 (26.5%) 21 (30.9%) 29 (49.2%)	*p* = 0.245
Q3. Sexual wellbeing <40 41–70 >71	37 (30.1%) 46 (37.4%) 40 (32.5%)	31 (45.6%) 19 (27.9%) 18 (26.5%)	*p* = 0.108
Q4. Physical wellbeing: chest <40 41–70 >71	77 (62.6%) 38 (30.9%) 8 (6.5%)	35 (51.5%) 15 (22.1%) 18 (26.5%)	*p* = 0.001
Breast sensitivity Preserved Lost	49 (39.8%) 74 (60.2%)	22 (32.4%) 46 (67.6%)	*p* = 0.350
Evaluation in patients with loss of breast sensitivity **			
	G1 76 (61.7%)	G2 44 (63.7%)	
Percentage of breast sensitivity lost 10–30% 40–70% 70–100%	35 (46.1%) 25 (32.9%) 16 (21.1%)	23 (52.3%) 13 (29.5%) 8 (18.2%)	*p* = 0.825
Influence on daily life 0–30% 40–70% 70–100%	35 (46.1%) 25 (32.9%) 16 (21.1%)	23 (52.3%) 13 (29.5%) 8 (18.2%)	*p* = 0.825
Influence on sexual life 0–30% 40–70% 70–100%	23 (30.3%) 23 (30.3%) 30 (39.5%)	17 (38.6%) 13 (29.5%) 14 (31.8%)	*p* = 0.607

* Evaluation performed on 195 (76.5%) patients who answered the questionnaire; ** evaluation performed on patients who experienced a loss of sensation after surgery. BCS: breast-conserving surgery; OPS: level II oncoplastic surgery; CMR: conservative mastectomy with reconstruction.

## Data Availability

Not applicable.

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
