# Peer review of "An Innovative Scoring System to Select the Optimal Surgery in Breast Cancer after Neoadjuvant Chemotherapy"

_jpm, 2023, doi:10.3390/jpm13081280_

Round 1

Reviewer 1 Report

Very interesting effort as the selection of surgery post neoadjuvant chemotherapy is still debatable. I would like to try it on patients in my clinic!

1.The aim of this study was to develop  a scoring system to choose surgery after neoadjuvant therapy in breast cancer patients, optimizing oncological and aesthetical outcomes.  2.It s an innovative attempt, addressing a gap of the literature, since selection of surgery was entirely in surgeon's convenience and personal taste. 3. A scoring system is easy to use,open tool that can help in decision making,comparing to other bibliographic references.  4.I cannot find any fault in methodology,the statistics seems adequate and the sample of patients is big enough. 5.The conclusions are consistent with the evidence and arguments presented. 6.No futher amelioration of figures and tables needed.  

I have no comments aw for english language.

Author Response

The authors thank for the review

Reviewer 2 Report

Introduction:

1.     ‘Factors associated with NACT include high tumor-to-breast volume ratio; lymph node-positive and specific biological features of primary cancer as high-grade disease; hormone receptor negative breast cancer (BC); triple negative (TN)[1]and overexpression of human epidermal growth factor receptor 2 (HER2+) phenotypes, even at an early stage (EBC).’ This sentence is not adequate. First, ‘factors associated with NACT’ is not an adequate phrase. You can write ‘Patients that will most probably benefit from NACT have the risk factors of …’. Besides, please give references for every factor. Not all patients with these factors give a good response to NACT.

2.     NACT does not aim to decrease the rates of axillary dissection (AD), please check your references 5 and 6 and the related sentence. We still have no prospective randomized trials reporting AD is not needed in patients undergoing NACT. And if only sentinel lymph node biopsy (SLNB) was performed, radiotherapy (RT) is recommended to the whole axilla which also has its complications. So, please revise that sentence.

Materials and Methods:

3.     I don’t think this is necessary: ‘Study population included patients undergoing three kind of surgery: 93 BCS, OPS and CMR. We excluded patients with: 1. lack of initial staging or restaging; 2. 94 previous or synchronous history of systemic malignant neoplasms. 3. history of homo-or 95 contralateral breast cancer. 4. initial evidence of metastatic pathology or development of 96 metastases during NACT; 5. radical mastectomy without reconstruction.’ as you already wrote it in figure 1. Please omit these sentences.

4.     ‘The indication to NACT was decided by a multidisciplinary team (MDT) composed by breast and plastic surgeons …’ Please revise ‘composed by’ as ‘composed of’.

5.     In 2.1; ‘Patients included in the study underwent three types of surgery: BCS, OPS and CM.’ Please revise ‘CM’ as ‘CMR’.

6.     The axillary surgery methods were written in two separate paragraphs. Please erase one of them (Under 2.1.; 2nd and 3rd paragraphs)

7.     Fig. 2; in the box below ‘Neoadjuvant Chemotherapy’, please revise ‘Locoregional ristaging’ as ‘Locoregional restaging’.

8.     2.4; why did you choose the DFS and OS rates start from the beginning of NACT? As a statistical method, survival rates are calculated from the date of diagnosis or the end of treatment so the efficacy of treatment can be measured. Your method of calculation should have increased your survival rates. Therefore, I suggest you also add the LR rates.

Results:

9.     Table 1; what do you mean by the numbers in the menopausal status row? Do the numbers mean the number of postmenopausal patients? Please revise.

10.  Table 2; the number of patients with microcalcification extension >8 mm is missing under the column of CMR. Please add.

11.  Please erase the duplication of information from the text that are already in Tables 1 and 2. Many numbers in parentheses make the text complicated and harder to understand. We already see them in Tables.

12.  3.4; The first paragraph is about the details of RT and Table S2 was referred. But Table S2 includes details about surgery and oncologic outcomes. Please check that.

13.  3.6.1, 3.6.2, 3.6.3; Please erase the p values in parentheses as we already see them in the tables.

14.  Table 4; 164 patients with G1 and 91 patients with G2. But when G2 patients in the oncological radicality column is summed, it makes 95 patients. There are also other incompatibilities with the numbers. Please check the numbers. Besides, what Table 4 is telling us is unclear. What is the purpose of Table 4 and what should we understand from it? Please describe in details. Also, if I understood right, should the numbers in the rows and columns not be switched?

15.  3.7; The last sentence of paragraph 3: ‘RL-DFS was 98.8% for G1 and 88.9% for G2…’ Please revise RL-DFS as LR-DFS.

16.  I have some questions and concerns for the Results section. Did the authors take into account the impact of RT and systemic treatment on the outcomes, particularly RT on the aesthetic outcomes? It is well known that RT may impair the aesthetic results, particularly in patients with reconstruction surgery. The rate of RT was different among the three groups. I think the authors should also add that into univariate analyses.

17.  Another question; how many patients with recurrence or bad aesthetic results undergo a second surgery and other additional treatments such as RT? Please state that for each group as it may have also affected the outcomes.

Discussion:

18.  This part is very weak. I would expect the authors to discuss about the impact of the variables in their UVA on the oncologic and aesthetic outcomes of breast cancer patients. Besides, there are many limitations to this study but only a few was mentioned. Among the most important limitations is the absence of RT in UVA and the impact of salvage surgery and other treatments in patients with LRR. With these deficiencies, I do not think this scoring system is reliable.

Some minor changes are required.

Author Response

According to your suggestion, we are resubmitting a new, revised version of the manuscript 

We believe that the manuscript is now improved and hope that it can be considered for publication

Reviewer 3 Report

Dear Authors,

The authors in manuscript entitled “An Innovative Scoring System to select the optimal surgery in 2 breast cancer after Neoadjuvant Chemotherapy” strongly suggested about selection of the breast surgery, optimizing oncological after the post – Neoadjuvant chemotherapy which is based on the post-neoadjuvant score system.

 Strengths of the study:

- This manuscript has written in descriptive manner with graphs.

- The reference list is not updated, need to add some recent references.

- The research article has concluded that the post-neoadjuvant scoring system based on clinical and radiological characteristics created to choose appropriate surgical treatment for breast cancer patients after NACT. Author described that this score tool system allows to reduce the risk of local reoccurrence and overall improving the patient’s quality of life. It also uses especially for tumors which do not shrink optimally.

-Author reported that no specific tool supports the surgeon in choosing surgery to date. Surgeons are selected the surgical technique is usually based on tumor characteristics (histotype, size and location), extent of resection, breast features (volumes, shape, glandular density), previous surgery, wishes of patients and surgeon’s expertise, presence of multifocality, extensive microcalcifications and a lobular histotype.

-Author claimed that there is no author that generated a predictive score system to select the optimal surgical procedure (BCS, OPS and CMR) based on pre and post NACT characteristics. Therefore, author’s aim was to design a standardized scoring system that can indicate the most appropriate surgical treatment to optimize oncological aesthetic outcome in breast cancer patients after NACT.

-For this study, author evaluated genetic, anatomical and radiologic features as BRCA mutations, breast ptosis, breast volume, pre and post NACT multifocality and multicentricity, pre-NACT MRI-PEBA, post-NACT RX-PEBA.

-Author claimed that pNESSy could facilitate the achievement of tumor-free margins and minimize the risk of local recurrence as regard oncological safety.

-Author also claimed that pNESSY could help surgeons and patients to choose the most performing type of surgery between BCS, OPS and CMR without compromising oncological safety as regards aesthetic outcomes and patient’s quality of life.

-Author claimed that no study described a model able not only to minimize recurrences, but also to optimize aesthetic outcomes. pNESSy constitutes the first standardized tool to determine the optimal surgery after NACT by taking into account clinical, radiological and genetics parameters. 

There are some issues with this article, if these issues are going to resolve then the quality of the paper is suitable for publication.

1)                  Introduction is well defined, there is need to write crisp.

2)                  Discussion should be elaborated with recent references.

3)                  There are few typos and English grammar errors which should be rectify.

4)                  Every part should be crisp, concise and well structured.

5)                  Conclusion prospective is clear.

6)                  Include the data collection format in supplementary.

Author Response

(The authors gave the same response as above.)
